# TEXT-TO-ENERGY: ACCELERATING QUANTUM CHEMISTRY CALCULATIONS THROUGH ENHANCED TEXT-TO-VECTOR ENCODING AND ORBITAL-AWARE MULTILAYER PERCEPTRON

## ABSTRACT

Accurately predicting material properties remains a complex and computationally intensive task. In this work, we introduce TEXT-TO-ENERGY (T2E), a novel approach combining text-to-vector encoding and a multilayer perceptron (MLP) for rapid and precise energy predictions. T2E begins by converting pivotal material attributes to a vector representation, followed by the utilization of an MLP block incorporating significant physical data. This novel integration of textual, physical, and quantum insights enables T2E to swiftly and accurately predict the total energy of material systems. The proposed methodology marks a significant departure from conventional computational techniques, offering a reduction in computational burden, which is imposed by particle count and their interactions, obviating the need for extensive quantum chemistry expertise. Comprehensive validation across a diverse range of atoms and molecules affirms the superior performance of T2E over state-of-the-art solutions such as DFT, FERMINET, and PSIFORMER.

## 1 INTRODUCTION

The exploration of electronic structures of molecules and solids in quantum mechanics is fundamentally guided by Density Functional Theory (DFT) (Hohenberg & Kohn, 1964; Parr & Yang, 1989; Jones & Gunnarsson, 1989; Dreizler & Gross, 1990; Martin, 2004; Saal et al., 2013), a robust computational method that anchors its principles on the resolution of the Schrödinger equation (Schrödinger, 1926; Ballentine, 1998; Atkins & Friedman, 2005; Griffiths, 2005; Cohen-Tannoudji et al., 2006), a fundamental equation describing the behavior of quantum particles within a potential. Despite its extensive utilization and significance, the application of DFT is not without its intricacies (Burke, 2012; Rudberg, 2012; Verma & Truhlar, 2020). Central to these is the challenge posed by the exchange-correlation (XC) functional, a critical term derived from the electron-electron interaction in the total energy expression of the system (Becke, 1988). This functional, expressed as:

$$\Delta E_{\text{XC}} = E_{\text{XC}}[\rho] - E_X[\rho] - E_C[\rho] \tag{1}$$

signifies the discrepancy or error in the XC functional, underscoring the difficulty in accurately capturing both exchange and correlation effects within a single functional, given its unknown exact form. This limitation manifests in the restricted description of various chemical phenomena, such as weak interactions and excited states (Cohen et al., 2008). To alleviate this, the Kohn-Sham offers an approximate solution by mapping the interacting system to a non-interacting one, thereby simplifying the many-body problem to a set of single-electron equations (Kohn & Sham, 1965). Nevertheless, the success of this method remains confined by the chosen XC functional. In response to these persistent issues, the adoption of machine learning (ML) (Alpaydin, 2020) techniques has emerged as a prominent strategy to augment the capabilities of traditional DFT methods (Ward et al., 2016; Brockherde et al., 2017; Schütt et al., 2017a; Zhang et al., 2018; Sahu et al., 2018; Xie & Grossman, 2018; Butler et al., 2018; Faber et al., 2019). ML techniques, particularly neural networks, have demonstrated substantial success in approximating complex wave functions (Cai &

Liu, 2018; Hermann et al., 2020), directly predicting observables (Hansen et al., 2015; Schütt et al., 2017b), and determining potential energy surfaces (Behler & Parrinello, 2007; Li et al., 2015), directly identifying functionals of DFT (Snyder et al., 2012; 2013; Li et al., 2016) and predicting a wide array of material forms (Fiedler et al., 2023).

Utilizing ML for predicting material properties faces two significant challenges. The *Computational Cost* is a foremost hurdle as the inherent complexity of many-body systems substantially prolongs the training duration, extending it from hours to weeks even with the handling of millions of parameters (von Glehn et al., 2022). Another crucial barrier is the *Requirement of Substantial Domain Knowledge*. Despite the abundant data available for energy estimation, the diversity in inputs that various ML strategies employ underlines the essential need for extensive domain knowledge (Pfau et al., 2020; Fiedler et al., 2023). This situation is further complicated by the reliance on pre-trained networks and scientific libraries such as PYSCF (Sun et al., 2018), adding another layer of approximation and complexity to the problem.

To address these issues, we introduce TEXT-TO-ENERGY (T2E), a novel approach that leverages text-to-vector encoding with a physics-informed multi-layer perceptron (PIMLP) for swift and accurate energy predictions. The text encoder alleviates the need for extensive domain knowledge by accepting material's name, spin, and 3D coordinates as input and converting them into a vector representation. This vector then serves as input for our PIMLP model, tailored for enhanced awareness of orbital characteristics, facilitating accurate material property predictions. Our PIMLP model, consisting of approximately $\approx 52K$ parameters, permits rapid energy predictions for unseen materials, representing a considerable advancement in computational efficiency compared to traditional *ab-initio* approaches, as delineated in Figure 1.

The structure of the remainder of the paper is outlined as follows: In section 2, we provide relevant background on Quantum Mechanics of many-particle systems and Kohn-Sham Functional Theory. Section 3 describes our T2E approach, while section 4 details our experiments for comparing our solution, T2E, with the DFT method and the state-of-the-art ML solutions FERMINET and PSI-FORMER. Finally, section 5 presents our conclusions.

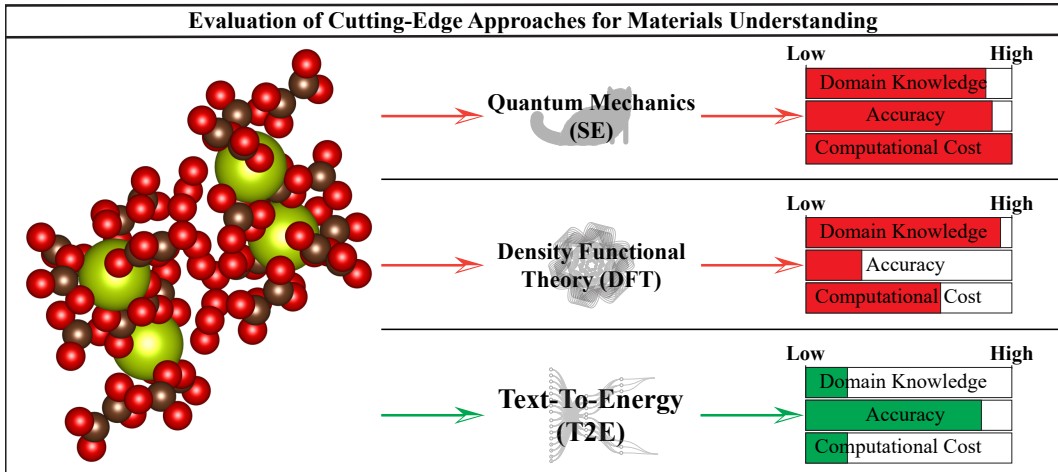

Figure 1: **Comparison of *ab-initio* approaches and T2E.** This figure underscores the trade-offs among cutting-edge material understanding methods. SE stands out for unmatched accuracy, at the cost of high computational demand due to the absence of fundamental approximations. DFT lessens computational load but sacrifices accuracy owing to approximation use. In contrast, T2E optimally blends efficiency and accuracy, demanding minimal domain knowledge and computational expense while offering accuracy equivalent to SE and surpassing the efficiency of DFT.

## 2 RELATED WORK

### 2.1 SCHRÖDINGER EQUATION

Understanding the behavior of many-particle systems, such as electrons in an atom or molecules, requires solving the many-body SE (Shankar, 1994). This equation serves as a cornerstone in quantum chemistry and condensed matter physics, offering a comprehensive depiction of the quantum state of systems comprised of multiple particles (Bruus & Flensberg, 2004). The time-independent SE for a system of N electrons in Dirac notation is given by:

$$\hat{H}\psi(\boldsymbol{r}_1, \ldots, \boldsymbol{r}_N) = E\psi(\boldsymbol{r}_1, \ldots, \boldsymbol{r}_N) \tag{2}$$

Here, $\hat{H}$ represents the Hamiltonian operator, which, for a Coulombic system of N electrons, can be expressed as:

$$\hat{H} = -\sum_i \frac{1}{2}\nabla_i^2 - \sum_{iA} \frac{Z_A}{|\boldsymbol{r}_i - \boldsymbol{r}_A|} + \sum_{i>j} \frac{1}{|\boldsymbol{r}_{ij}|} \tag{3}$$

Breaking down this Hamiltonian, the first term encapsulates the kinetic energy of each electron, and the second term accounts for the potential energy experienced by each electron within the external potential. The third term characterizes the Coulombic interactions between pairs of electrons (Jackson, 1999), with the summation over $i$ and $j$ encompassing all pairs within the system. Solving the time-independent SE (eq. 2) for a system of N electrons presents a formidable challenge, as exact solutions for Coulombic many-electron systems are generally analytically unattainable, except for a few special cases. Consequently, researchers employ various approximations and numerical techniques, such as DFT and Variational Monte Carlo (VMC) (Foulkes et al., 2001), to surmount these challenges and obtain approximate solutions.

### 2.2 KOHN-SHAM DENSITY FUNCTIONAL THEORY

Solving the many-body SE demands substantial computational resources, and as the system size increases, the demands for memory and processing power grow exponentially, making the computational cost prohibitive for large systems (Ratcliff et al., 2017). However, Kohn-Sham (KS) DFT (Kohn & Sham, 1965) emerges as a powerful solution to address these challenges (Becke, 2014). DFT is a groundbreaking approach that simplifies the task of solving the many-body SE while maintaining accuracy. Within the DFT framework, the Kohn-Sham formula provides a straightforward expression for the ground state energy, which can be represented as the sum of several terms:

$$E[\rho] = T_s[\rho] + V_{ext}[\rho] + E_H[\rho] + E_{xc}[\rho] \tag{4}$$

where $T_s[\rho]$ is the kinetic energy term, which represents the total kinetic energy of the electrons for the KS noninteracting reference system, which can be given as:

$$T_s[\rho] = \sum <\phi_i| - \frac{1}{2}\nabla^2|\phi_i> \tag{5}$$

in terms of $\phi_i$, the set of one electron KS orbitals. The electron density (ED) of the KS reference system is given by:

$$\rho(\boldsymbol{r}) = \sum |\phi_i(\boldsymbol{r})|^2 \tag{6}$$

The $V_{ext}[\rho]$ term in Eq. 4 accounts for the potential energy of the electrons due to the presence of external fields or nuclei. This potential term can be expressed as:

$$V_{ext}[\rho] = \int \rho(\boldsymbol{r})v(\boldsymbol{r})d\boldsymbol{r} \tag{7}$$

The second-to-last term, $E_H[\rho]$, is the Hartree energy term, which represents the classical electrostatic interaction (electron - electron repulsion) energy between the electrons. This term can be written as:

$$E_H[\rho] = \frac{1}{2} \int \int \frac{\rho(\boldsymbol{r})\rho(\boldsymbol{r}')}{|\boldsymbol{r} - \boldsymbol{r}'|} d\boldsymbol{r} d\boldsymbol{r}' \tag{8}$$

The last term is the exchange-correlation energy term, which accounts for the quantum mechanical exchange and correlation effects between the electrons. It can be expressed in the constrained search formulation for density functionals as follows, although no explicit form is available (Levy, 1979).

$$E_{XC}[\rho] = (T[\rho] - T_s[\rho]) + (V_{ee}[\rho] - E_H[\rho]) \tag{9}$$

It's important to note that the choice of the exchange-correlation functional can have a significant impact on the accuracy of DFT calculations for different systems and properties. Different functionals may perform better or worse depending on the specific problem at hand. While the KS equations provide a powerful framework for approximating the electronic structure of a system within DFT, there are still several challenges and limitations associated with their practical implementation. The followings are some of the key challenges in the KS equations: Exchange-correlation approximation (Von Barth & Hedin, 1972; Perdew & Yue, 1986; Arbuznikov, 2007); band gap problem (Perdew, 1985; Perdew et al., 2017); treatment of strongly correlated system (Avella et al., 2012); self-consistency (Wasserman et al., 2017); sensitivity to basis set (Fouda & Besley, 2018); system size (Pan, 2021).

## 3 T2E: TEXT-TO-ENERGY

In this section, we present our T2E model, composed of a text-to-vector encoder and a PIMLP model. Together, they address significant challenges. The encoder minimizes the necessity for extensive domain knowledge by excluding particle and interaction calculations from complexity assessment, thereby enhancing scalability. Simultaneously, our PIMLP model efficiently manages the balance between the need of physical information for generalization and extensive computational requirement.

### 3.1 TEXT-TO-VECTOR ENCODER

Theoretical material simulations demand extensive domain knowledge. Even with packages like PYSCF that aim to mitigate this need, detailed inputs like basis vectors and material coordinates are crucial, with output precision heavily reliant on these parameters. ML models like FERMINET and PSIFORMER also lean on PYSCF for initial settings, maintaining the necessity for significant domain insight. Table 1 contrasts the settings of the state-of-the-art models with T2E, highlighting the maintained demand for extensive knowledge in other models. Our T2E model employs a text-to-vector encoder, significantly diminishing the extensive domain knowledge typically required by reducing necessary inputs to merely the material's name, coordinates, and spin. This encoder, illustrated in detail in Figure 2, involves four layers: a tokenizer, a positional embedder, a transformer (Vaswani et al., 2017), and a layer normalization (Ba et al., 2016) layer, working sequentially to convert text inputs into a normalized vector representation. This vector is then utilized by the PIMLP module. T2E's encoder efficiently calculates attention scores, $\alpha$, leveraging a self-attention mechanism for identifying and weighting relevant queries within a sequence, enhancing the model's robustness and precision. $\alpha = p_{\pi,\tau}(\mathbf{Q}\mathbf{K}^T \times \sqrt{d_k}^{-1}) \cdot \mathbf{V}$, where $\alpha$ is the attention weights, $p_{\pi,\tau}(\cdot)$ is the softmax function with temperature $\tau$ and permutation $\pi$, $\mathbf{Q}$ is the query matrix, $\mathbf{K}$ is the key matrix, $\mathbf{V}$ is the value matrix, and $d_k$ is the dimension of the key vectors. In Transformers, position-wise feed-forward networks are applied to the output of the attention mechanism as follows: $\text{FFN}(X) = \text{ReLU}(XW_1 + b_1)W_2 + b_2$. Here, $X$ denotes the attention output, with $W_1, b_1, W_2, b_2$ representing the learnable weights and biases. These equations form the crux of the Transformer's attention mechanism, crucial for discerning and capturing token dependencies and relationships in a sequence, allowing the model to contextually weigh word significance. We utilize pretrained CLIP as text embedder (Radford et al., 2021), a neural network adept at converting

Table 1: **Comparison of required setting characteristics.** T2E notably reduces the necessity for extensive domain knowledge compared to other models, promoting easier accessibility and utilization in the field while ensuring maintained structure and minimal input dimensions.

| | Settings | | | | | | | |
|---|---|---|---|---|---|---|---|---|
| | **Material** | **Coordinates** | **Spin** | **Basis** | **Initial Guess** | **XC** | **Grid Settings** | **Optimizer** |
| **PySCF** | Required | Required | Required | Required | Required | Required | Required | Required |
| **FermiNet** | Required | Required | Required | Uses PySCF | Uses PySCF | Uses PySCF | Uses PySCF | Itself + Uses PySCF |
| **PsiFormer** | Required | Required | Required | Uses PySCF | Uses PySCF | Uses PySCF | Uses PySCF | Itself + Uses PySCF |
| **T2E** | Required | Required | Required | Not Used | Not Used | Not Used | Not Used | Itself |

text and visuals into significant vectors using contrastive learning. This type of learning starts with calculating cosine similarity score ($S$) between text ($t_i$) and image embeddings ($x_i$) as follows to evaluate the effectiveness of the embeddings.

$$S(t_i, x_i) = \frac{\widetilde{t_i} \cdot \widetilde{x_i}}{\|\widetilde{t_i}\| \cdot \|\widetilde{x_i}\|} \tag{10}$$

where, $\widetilde{t_i}$ represents the normalized text embedding, $\widetilde{x_i}$ represents the normalized image embedding, and $\|\cdot\|$ denotes the Euclidean norm used for normalization. Subsequently, the contrastive loss ($L$) for a positive pair and a negative pair can be defined as:

$$L(t_i, x_i) = -\log\left(\frac{\exp(S(t_i, x_i)/\tau)}{\sum_{j=1}^{N} \exp(S(t_i, x_j)/\tau)}\right) \tag{11}$$

In Equation 11, $t_i$ and $\tau$ represents the embedding and temperature parameter, respectively, $N$ is the total number of negatives (unrelated text-image pairs), and $\cdot$ denotes the dot product.

In our work, we leverage the text encoder part of this powerful tool to lower the barrier of computation and domain knowledge required for material property predictions. We achieve this by using the desired material's name, 3D coordinates, and spin value as input to the encoder. The encoder then instantly converts this information into a 768-dimensional vector ($x_{material}$). Different vectors are obtained for different materials, coordinates, and spin combinations, depending on the input. This process is illustrated in Fig. 2. For the shape of the carbon atom, we use the VESTA package (Momma & Izumi, 2008). After obtaining the vector representation of the text, it is then fed into our MLP layer, where the model is provided with reminders of physical and quantum properties at various stages.

## 3.2 ORBITAL-AWARE PIMLP

The MLP component of T2E is physics-informed (Karniadakis et al., 2021), proficiently extracting physical relations from data using the quantum property, spin (Griffiths, 2020). Spin (**S**), vital for explaining various phenomena, is used to describe the angular properties and behavior of the particle system (Atkins & Friedman, 2005). Total atomic spin is fundamentally calculated after determining each particle's spin. Using the Dirac Equation for electron (Dirac, 1926; Sakurai & Commins, 1995):

$$(i\hbar\boldsymbol{\gamma}^\mu \partial_\mu - mc)\psi(x, t) = 0 \tag{12}$$

where $\psi(x, t)$ is the Dirac spinor, a four-component wavefunction that describes the electron. $\boldsymbol{\gamma}$ is a set of $4x4$ complex matrices, $m$ is the rest mass of electron, and $c$ is the speed of light. A solution for a resting electron is $\psi(x, t) = (1\ 0\ 0\ 0)$. The $z$ component of the spin is $\mathbf{S_z} = \frac{1}{2}\hbar(\mathbf{1} - \boldsymbol{\beta})$, where $\hbar$ is the reduced Planck's constant and $\boldsymbol{\beta}$ is a Dirac matrix. We can calculate the eigenvalues and eigenvectors associated with $\mathbf{S_z}$ using the following equation:

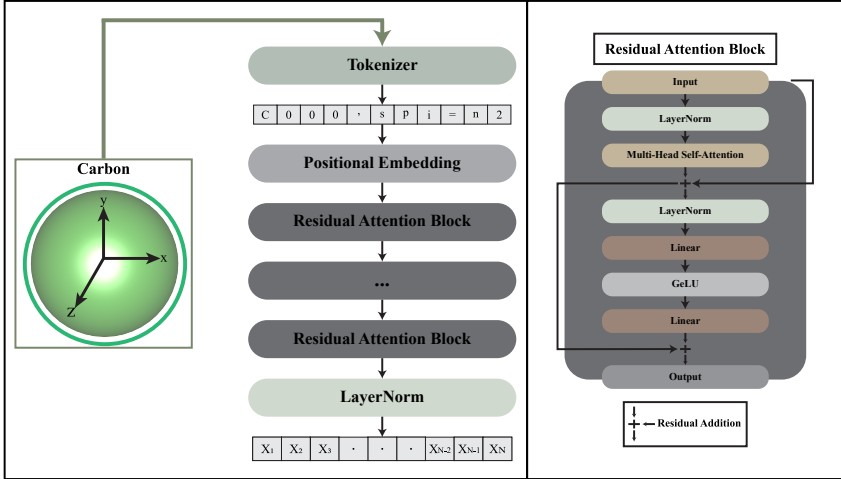

Figure 2: **Text embedding process of T2E.** Material details are first tokenized and positionally embedded, preparing them for the transformer's residual attention block. Following layer normalization, a 768-dimensional vector representation is obtained, ready for input into the orbital-aware, PIMLP of T2E.

$$\mathbf{S_z} \left| \psi(x,t) \right\rangle = s \left| \psi(x,t) \right\rangle \tag{13}$$

yielding $s = -\frac{\hbar}{2}$, the electron's spin. A similar process for protons and neutrons yields spin values of $\pm\frac{\hbar}{2}$. The $\pm$ represents the direction while $1/2$ is the magnitude of spin vector in natural units. The atomic spin, a sum of individual particle spins, significantly influences system properties. Known spins of particles contribute to the total spin $I_{\text{nuclear}} + I_{\text{electronic}}$. In our model, this information is encoded in an MLP layer. The distinctive mass number $A$ (sum of proton ($Z$) and neutron ($N$) numbers) provides $I_{\text{nuclear}}$ information, and, in charge-neutral systems where $Z = e_{\text{count}}$, $I_{\text{electronic}}$ information is also obtained. Furthermore, valence electrons ($e_{\text{valence}}$) dictate chemical properties, bonds, and atomic interactions (Kittel & McEuen, 2018), carrying significant information about the sysmtem (Szabo & Ostlund, 2012). Therefore, we incorporate $e_{\text{valence}}$ to the MLP along with mass number. The additional features in our network can be expressed as $x = [x_{material}, A, e_{\text{valence}}]$. Once the vector representation of the input is acquired from T2E's embedder, it is fed into the initial layer of our orbital-aware PIMLP. During the information processing within the network, it passes through ReLU activation functions (Fukushima, 1975) after each linear layer except the last one, integrating spin and mass number information to extract physical data. Refer to Figure 3 for an illustration of the predictive part of T2E. For a given input vector $x$, the forward pass of an MLP is computed as:

$$a^{(1)} = x, \qquad z^{(i)} = W^{(i)}a^{(i)} + b^{(i)}, \qquad a^{(i+1)} = f(z^{(i)}) \tag{14}$$

where $a^{(i)}$ is the activation of layer $i$, $z^{(i)}$ the weighted sum of layer $i$, $W^{(i)}$ the weight matrix, $b^{(i)}$ the bias vector, and $f$ the applied element-wise activation function.

## 4 EXPERIMENTS

We generated simulated data using PYSCF, obtaining $\approx 6000$ data for each benchmarked element and molecule by altering coordinates, which correspond to different text-embedding and total energy, excluding non-converged positions. Our model, built with PYTORCH (Paszke et al., 2017), follows an 80/20 train-validation split with batch size of 64, utilizing ADAM (Kingma & Ba, 2014) for optimization, and mean squared error (MSE) for training loss. Additionally, implementing weight decay, dropout, and vector augmentation for overfitting mitigation. We evaluated T2E on first and

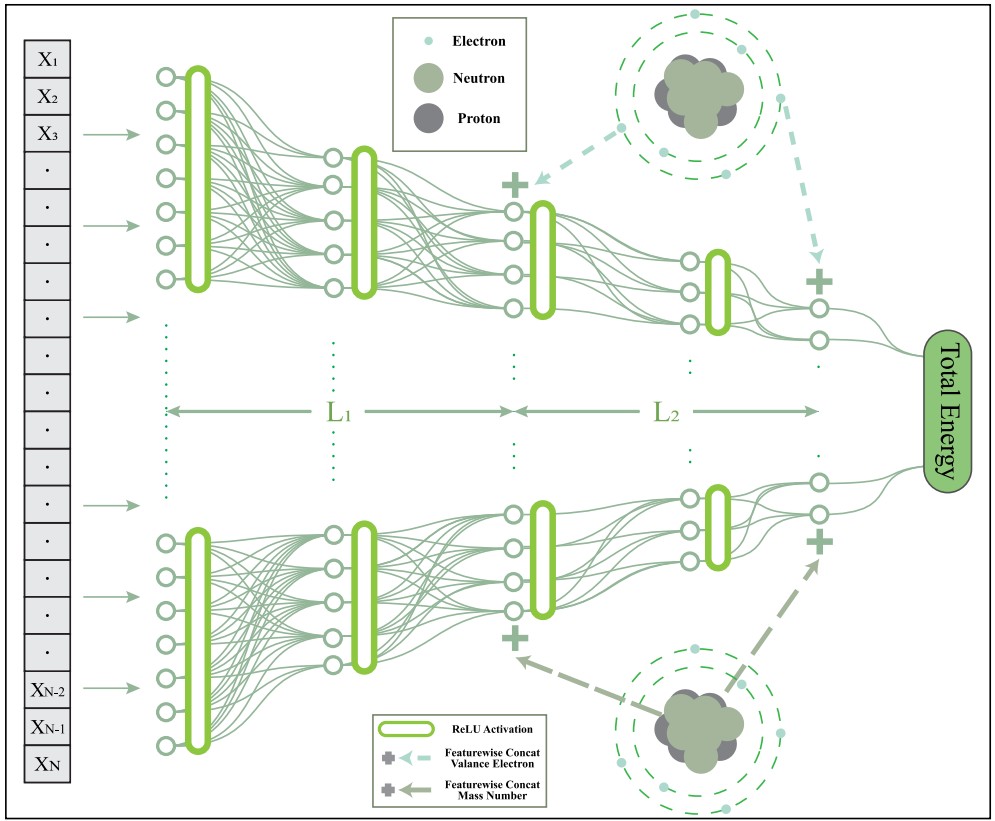

Figure 3: **Orbital-aware PIMLP in T2E.** The orbital-aware PIMLP of T2E predicts total energy from a 768-dimensional vector produced by the text-encoder, outputting a single variable for the energy prediction of the input material. Physical and spin informations are reminded to the network at the connection of $L_1$ and $L_2$ and at the end of the $L_2$ just before the prediction.

second-row elements (excluding He, F, and Ne), untrained elements, and Li——H, Li——Li, C≡≡≡O, and N≡≡≡N molecules. Section 4.1 and 4.2 detail DFT experiments via PYSCF with *ccpvdz* basis, FERMINET, and PSIFORMER implementations. Training continued until mean variance was $< 0.001$ and *pmove* was $< 0.5$ while VMC values from Pfau et al. (2020). Section 4.3 maintains PYSCF DFT, securing FERMINET and PSIFORMER values from von Glehn et al. (2022) and reference value of C≡≡≡O is from (Powell & Dawes, 2016). In-house simulations and experiments were conducted with an RTX4070 Laptop GPU. More details about simulations and experiments are given in supplementary material.

## 4.1 FIRST AND SECOND ROW ELEMENTS

Initially, we embark on a standard experiment in the literature. This experiment involves assessing the performance of the proposed approach on various first and second-row periodic table elements: H, Li, Be, B, C, N, and O. The outcomes of this experiment are delineated in Table 2. T2E outperforms benchmarked models, showing superior precision to DFT and equivalent accuracy with FERMINET, PSIFORMER, and VMC within a $1\%$ precision range. Uniquely, T2E achieves this without prior training or substantial domain knowledge, unlike FERMINET and PSIFORMER which start optimizing with pretrained values from PYSCF, leveraging only periodic table information to yield comparable results to those demanding more complex prerequisites. Additionally, while the precision of VMC is highly contingent upon its trial function, necessitating significant domain knowledge and an expected ground state shape, T2E consistently delivers closely aligned results using merely periodic table information.

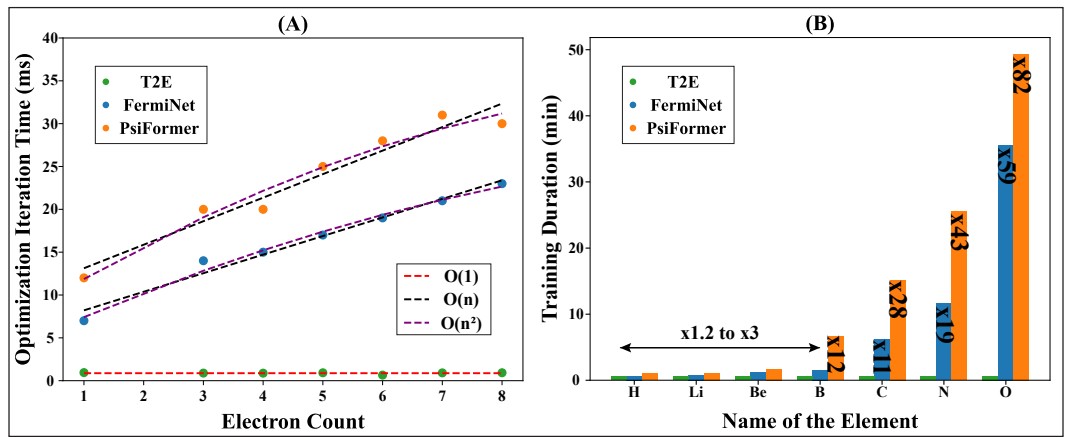

Figure 4: **Optimization and training time comparison.** (A): Single optimization time and scaling type for each model, determined by polynomial fitting. (B): Training time for each model to approach $\pm 0.01\%$ of the reference, both conducted on a RTX4070 laptop.

Table 2: **Total energy predictions for first and second row elements (H to O), excluding He:** This table presents total energy values for DFT, FERMINET, and PSIFORMER, derived from our simulations, alongside VMC and reference values from existing literature. All energy values are denoted in Hartrees. Pertinent references are cited within the main text, and a comprehensive precision analysis is available in the supplementary material.

| ELEMENT | REFERENCE | DFT | T2E | FERMINET | PSIFORMER | VMC |
|---|---|---|---|---|---|---|
| Hydrogen (H) | $-0.5$ | $-0.5$ | $-0.5$ | $-0.5$ | $-0.5$ | NA |
| Lithium (Li) | $-7.5$ | $-7.3$ | $-7.4$ | $-7.5$ | $-7.5$ | $-7.5$ |
| Beryllium (Be) | $-14.7$ | $-14.4$ | $-14.6$ | $-14.7$ | $-14.7$ | $-14.7$ |
| Boron (B) | $-24.7$ | $-24.3$ | $-24.5$ | $-24.6$ | $-24.6$ | $-24.7$ |
| Carbon (C) | $-37.8$ | $-37.4$ | $-37.7$ | $-37.8$ | $-37.8$ | $-37.8$ |
| Nitrogen (N) | $-54.6$ | $-54.1$ | $-54.4$ | $-54.6$ | $-54.6$ | $-54.6$ |
| Oxygen (O) | $-75.1$ | $-74.5$ | $-74.8$ | $-75.1$ | $-75.1$ | $-75.1$ |

The primary computational challenges arise from the size of the material. Larger materials entail a higher number of particles, leading to increased interactions and subsequently escalating both computational costs and parameter counts, especially in FERMINET and PSIFORMER. Despite these challenges, T2E offers a distinct advantage, as highlighted in Figure 4. Notably, for elements from H to O (excluding He), T2E exhibits constant scaling. In contrast, FERMINET and PSIFORMER demonstrate $\mathcal{O}(n)$ and $\mathcal{O}(n^2)$ scalings, respectively. This advantage arises because T2E employs a consistent input structure across all materials, characterized by an embedded text with 768 features. Furthermore, T2E surpasses traditional approaches in iteration time, credited to its minimal parameter count of ($\approx 52k$). As depicted in Figure 4 (B), when evaluating the training durations for FERMINET, PSIFORMER, and T2E, T2E remains unmatched. Specifically, as the electron count augments eight-fold from H to O, the training times soar approximately $80\times$ and $60\times$ for PsiFormer and FermiNet, respectively, juxtaposed with T2E. Unlike conventional methods that front-load physical knowledge, escalating computational demands, T2E incorporates this knowledge seamlessly via its orbital-informed MLP. Consequently, T2E accomplishes remarkable precision without compromising on computational efficiency.

## 4.2 PREDICTION OF NEW MATERIALS

In this experiment, we evaluate the capability of T2E in predicting the total energy of untrained elements. The model, trained exclusively on Li, Be, and C, attempts to forecast the total energy of B and trains exclusively on Li, Be, and B, attempts to predict the total energy of C. The detailed outcomes are delineated in Figure 5. Figure 5 (A) emphasizes T2E's proficiency in precisely

Table 3: **Total energy predictions for common molecules:** The table showcases total energy values from FERMINET and PSIFORMER, sourced from prior studies, and DFT values from PYSCF simulations. All energy values in Hartrees. Note: NC indicates non-convergence. See the supplementary material for an extensive precision analysis and details on bond lenghts.

| MOLECULE | REFERENCE | T2E | FERMINET | PSIFORMER | DFT |
|---|---|---|---|---|---|
| Li——H | −8.1 | −8.0 | −8.1 | −8.1 | −7.8 |
| Li——Li | −15.0 | −15.0 | −15.0 | −15.0 | −14.7 |
| C≡O | −112.9 | −113.7 | −113.3 | −113.3 | −102.1 (NC) |
| N≡N | −109.5 | −110.4 | −109.5 | −109.5 | −108.0 |

estimating the total energy of an unseen element, B, underscoring its capacity to internalize and generalize physical information from limited training data. This outcome further indicates that the incorporation of mass number and valence electron information enables the model to learn behavior of quantum systems. One can argue that the spin and valance electron values of B fall within the range of those for Li to C, leading to the expectation that the model would successfully predict the total energy of B. However, as demonstrated in Figure 5 (B), T2E continues to perform exceptionally even when tested on elements outside the training parameter range. This serves as further proof of T2E's ability to extract and generalize physical information using only basic periodic table parameters.

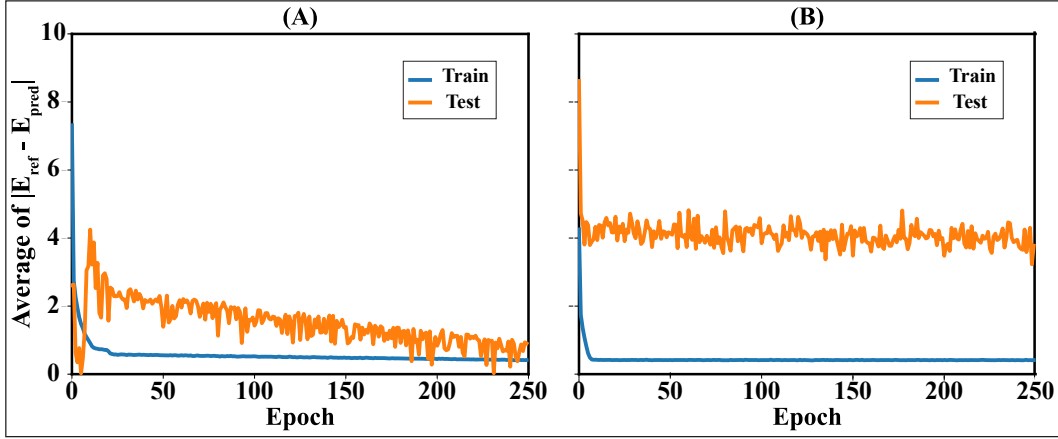

Figure 5: **Total energy prediction performance of T2E for new materials.** T2E's performance when its tested to predict total energy of untrained elements. Vector augmentation and weight decay kept same while lowering the batch size to one. (A): Prediction of total energy of B when trained on data of Li, Be, and C. (B): Total energy prediction of C when trained on data of Li, Be, and B.

### 4.3 MOLECULES

As a general experiment, we assess the performance of T2E on molecules of Li——H, Li——Li, C≡O, and N≡N. Observations from Table 3 indicate that T2E retains its competitive edge with the same number of parameters used. It exhibits superior accuracy compared to DFT, while remaining within a 1% range of the reference.

### 5 CONCLUSION AND FUTURE WORK

Our T2E model is a promising new tool for quantum property predictions. It outperforms state-of-the-art models such as DFT, FERMINET, PSIFORMER, and VMC on a variety of elements and molecules, without requiring prior training or extensive domain expertise. Future work will focus on expanding T2E to heavier elements and highly correlated systems, as well as exploring advanced quantum features.

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
