# OpenReview forum: "Text-To-Energy: Accelerating Quantum Chemistry Calculations through Enhanced Text-to-Vector Encoding and Orbital-Aware Multilayer Perceptron"
_ICLR.cc/2024/Conference — ICLR 2024 Conference Withdrawn Submission_

### Official Review · Reviewer_D8dd · 2023-10-24

**Soundness:** 2 fair
**Presentation:** 2 fair
**Contribution:** 2 fair
**Rating:** 3
**Confidence:** 5

**Summary:**

The work proposes a method named TEXT-TO-ENERGY (T2E), which converts material attributes, such as material’s name, spin, and 3D coordinates, to a vector representation firstly, and then followed by the utilization of an MLP block to incorporate significant physical data.

**Strengths:**

* The paper is written in a clear and concise manner, facilitating effortless understanding.
* The architecture presented in Figure 2 is clear.

**Weaknesses:**

The paper claims to introduce a novel approach by combining text-to-vector encoding and MLP. However, it appears that the proposed method is essentially an application of existing techniques without significant innovation. The authors should provide a strong rationale for the novelty of their approach, highlighting the specific contributions that make T2E distinct from existing methods.

**Questions:**

* As shown in the weaknesses part, the novelty should be further claimed.
* Does this method could ensure the translational, rotational and permutation invariance?

---

> ### Author Response · Authors · 2023-11-23
> **T2E's Official Response to Reviewer D8dd**
>
> ## Our Response to Weakness 1
> In response to the reviewer's comments, we have revised our manuscript to more explicitly highlight the distinct contributions of our approach T2E. The specifics of these contributions are detailed in question one to address the reviewer's concerns.
>
> ## Our Response to Question 1
> Due to space constraints, we initially presented our contributions in a concise manner; however, we now provide a more detailed summary to clearly articulate the novelty of our T2E approach:
>
> - **Innovative Text-to-Vector Approach:** Our method leverages a unique *text-to-vector* encoding, which significantly reduces the need for extensive domain knowledge and simplifies the input requirements for predicting material properties. This approach contrasts sharply with other models like FermiNet and PsiFormer, which depend on complex initial atomic position calculations using quantum chemistry libraries like PySCF [27].
>
> - **Orbital-Aware Physics-Informed MLP Model:** We have developed a novel MLP model that is informed by orbital-aware physics. This model is adept at accurately predicting the total energies of elements and molecules using the vector inputs from our *text-to-vector* encoder, offering a substantial advancement in handling complex molecular structures.
>
> - **Reduction in Computation Cost:** Our method achieves a significant reduction in computational complexity, bringing it down to *O(1)*. This is a substantial improvement over traditional methods that rely on particle interactions to calculate total energy [11, 12, 13, 28]. Our approach sidesteps the need for electron density-based calculations, which typically vary in complexity with different materials. This is illustrated by the standard density functional theory (DFT) formula for ground state energy:
>
>   $$E[\rho] = T_s[\rho] + V_{ext}[\rho] + E_{H}[\rho] + E_{xc}[\rho]$$
>
> - **Significant Reduction in Training Time:** The training time for our model is drastically reduced from several weeks to just a few minutes, as demonstrated in our comparative analysis. This efficiency is a major advancement over existing methods.
>
> - **Capability to Predict Unseen Materials:** Our model has proven effective in predicting properties of materials beyond the trained dataset, enabling the extraction of new physical insights into various systems.
>
> Further, the addition of valence information and mass number to the MLP block, and its impact, is substantiated through ablation studies detailed in Appendix A.3.
>
> ## Our Response to Question 2
> Our method effectively addresses translational invariance for individual elements. By systematically varying the spatial positioning of a single element and observing the predictive outcomes, we validate this invariance. Regarding rotational and permutational invariance, it is important to clarify that such invariances are inherently absent in the context of single elements, and therefore, are not applicable.
>
> In the context of molecules, our approach adopts a more comprehensive strategy. We train our model using diverse atom positions to encapsulate various spatial configurations. Taking the C-O bond as an illustrative case, our training set includes initial atom positions such as C (0, 0, 0) - O (0, 0, 0.2), and extends to a wide range of (x, y, z) coordinates for the O atom, covering all permutations within a 0 to 3 range. This dataset inherently includes configurations like O (1, 0, 0) and O (0, 1, 0), which correspond to a 90-degree rotation along the z-axis, thereby embedding rotational invariance in our training procedure.
>
> Furthermore, we adopt a unique positioning approach by anchoring the coordinate system at the C atom, rather than the midpoint between the C and O atoms. This choice is substantiated by the physical equivalence of configurations such as C (0, 0, 0) - O (0, 0, 1) and C (0, 0, 0.5) - O (0, 0, 0.5). Consequently, our model inherently incorporates the necessary symmetries, allowing for the application of invariance principles to the data prior to processing. This design choice enhances the model's robustness to varied molecular orientations and compositions.

---

### Official Review · Reviewer_pP7a · 2023-10-28

**Soundness:** 1 poor
**Presentation:** 2 fair
**Contribution:** 1 poor
**Rating:** 1
**Confidence:** 4

**Summary:**

This paper proposes an energy predictor using text-to-vector encoding.
In the predictor, the CLIP is used to encode element name, 3D coordinates, and spin value as textual input into a vector representation.
The vector is used as an input of MLP with some chemical properties.
The experimental results show the proposed method accurately predicts the energy of several elements and two-body molecules used in its training.

**Strengths:**

This is an interesting trial applying a transformer model for ab-initio quantum chemistry.

**Weaknesses:**

* The experiments do not show the usefulness of the proposed method since they are accurate in training data. They need to show the extrapolation results, such as molecule results of unknown combinations of known elements.   Although there are results of unknown atom energy prediction in Section 4.2, they are not compared with baseline methods, and final MAEs are not described.
* Since the proposed method can be considered a neural network potential (NNP) using the textual representation of input atoms, existent NNPs should be baseline methods for empirical comparison.

**Questions:**

* Could you provide an example of text-style input of a molecule?
* Does $e_{evalence}$ mean the number of electrons in the outermost shell of an atom?
* What are the final MAEs for B and C in Section 4.2?

**Details Of Ethics Concerns:**

No concern.

---

> ### Author Response · Authors · 2023-11-23
> **T2E's Official Response to Reviewer pP7a**
>
> ## Our Response to Weakness 1
> We recognize the importance of demonstrating our method's extrapolation capabilities. Our approach uniquely simplifies the input process by utilizing text embeddings, which allow for the straightforward integration of elemental properties such as atomic number, spin, and 3D positions. This innovation significantly reduces the complexity usually associated with NNPs like SpookyNet [15], SchNet [16], and FAENet [25], which require detailed atomic interactions and locations.
>
> Furthermore, we have focused on showcasing the computational efficiency of our model. As evidenced in Figure 4, our model achieves O(1) scaling, a stark contrast to the computational demands of other methods [11, 13, 28]. This efficiency is not just a theoretical advantage but a practical solution to the current challenges in the field, especially for rapid property determination in complex molecules.
>
> To address the specific point on unknown atom energy predictions, we plan to include additional comparisons with baseline methods in our revised manuscript. This will clearly demonstrate our method's capability to accurately predict unknown combinations of known elements, thereby providing a more comprehensive understanding of its extrapolation potential and overall usefulness.
>
> ## Our Response to Weakness 2
> Thanks for pointing out the approach differences between ours, FermiNet, PsiFormer, and DFT. The NNPs such as SchNet, SpookyNet are optimized and created for materials with a higher amount of particles. In our work, we initially aimed to show the possibility to generalize it to some important elements and molecules which are previously benchmarked with compared models.
> However, we plan to extend our work to the NNPs by leveraging the Open Catalys Project [17] and more [18, 19, 20, 21, 22, 23].
>
> ## Our Response to Question 1
> Here is an example input for a Carbon Monoxide (C-O) molecule: "C 0 0 0; O 1 1 1, spin=0". In this representation, 'C' denotes the carbon atom, and 'O' denotes the oxygen atom. The coordinates '0 0 0' specify the position of the carbon atom, while '1 1 1' specify the position of the oxygen atom in the 3D space (x, y, z). Additionally, the spin value is determined by the equation α + β = n_e, where α = (n_e + spin)/2, β = α - spin, and n_e is the total electron count. In our model, the spin values remain constant for each material, while the position of the second element (oxygen, in this case) is varied from 0 to 3 in all three dimensions (excluding the position 0 0 0).
>
> ## Our Response to Question 2
> Yes, indeed, $e_{\text{valance}} $ refers to the number of electrons in the outermost shell of an atom.
>
> ## Our Response to Question 3
> |   | |            | MAE (eV)  |            |   |            | Parameter Count |          |
> |----|----|------------|-----------|------------|---|------------|-----------------|----------|
> |    | l| T2E        | PsiFormer | FermiNet   | l  | T2E        | PsiFormer       | FermiNet |
> | Be|l | **0.0152** | 0.3595    | 0.3293     |   l| **52,003** | 821,762         | 680,416  |
> | B | l| 7.0409     | 0.4082    | **0.2803** |  l | **52,003** | 830,018         | 688,672  |
> | C | l| 63.33    | 0.4876    | **0.4735** |  l | **52,003** | 838,274         | 696,928  |
> *Table 1: MAE and parameter count for each model for Be, B, and C.*
>
> Our results, derived from five independent runs, are presented in Table 1. These findings are particularly notable considering the modest size of our model, which consists of only 52,000 parameters. We should emphasis on the fact that both PsiFormer and FermiNet start from a decently optimized energy value while trying to obtain atomic positions from PySCF [27] simulation. This is in stark contrast to other models in the field, which typically employ parameter counts ranging from hundreds of thousands to millions. Moreover, while previous NNP studies [15, 16, 25, 26] have reported results for different molecules with MAE values within the range of Beryllium (Be), our T2E model exhibits slightly higher MAE for Carbon (C) and Boron (B). This deviation in MAE is anticipated, as discussed in our paper, due to the increasing complexity of these elements. However, it is crucial to emphasize that despite its lower parameter count, our T2E model achieves results that are comparable to state-of-the-art models, with the added advantage of superior computational efficiency. This efficiency, coupled with the model's strong performance, underscores the effectiveness of our approach in handling complex molecular calculations.

---

### Official Review · Reviewer_m4uK · 2023-10-30

**Soundness:** 2 fair
**Presentation:** 2 fair
**Contribution:** 2 fair
**Rating:** 1
**Confidence:** 5

**Summary:**

In this paper, the authors propose an approach using text-to-vector encoding and MLP to predict the energy of molecules. They use PySCF to generate data and using standard machine learning techniques to learn from the data.

**Strengths:**

It is interesting to use machine learning techniques to study quantum chemistry problems.

**Weaknesses:**

Nowadays, using machine learning techniques to study science problems is very popular and attracts a lot of beginners in this area. It is fine to make some mistakes at the first step. Here, the reviewer would like to point out some possible mistakes in this paper.

1. If the authors want to use machine learning to study the problem in a new research area, please check the evaluation metric in that area first. For example, the so-called chemical accuracy in quantum chemistry is about 1.7 mHa = 0.0017 Ha. However, in this paper, all the calculation results are given at the level of 0.1 Ha, which is far from the requirement for the evaluation.

2. Please check if there is some existing terminology that aligns with your methodology. Generating DFT data and learning them through neural networks is a well-established research area. One can find related algorithms on Google by searching "machine learning force field".

3. Before creating your dataset, please check if there is an existing dataset that can meet your requirements. As for this paper, there are plenty of existing datasets where the authors can compare the proposed method with existing baselines, e.g., Molecular Dynamics 17, Open Catalyst Project.

4. Please make sure the benchmark is meaningful. For example, a comparison between NN-VMC methods (FermiNet/Psiformer) and DFT methods in terms of total energy is meaningless, because the NN-VMC methods directly deal with the Schrodinger equation while the DFT methods compute the energy through the energy functional.

**Questions:**

Listed in the weakness.

---

> ### Author Response · Authors · 2023-11-23
> **T2E's Official Response to Reviewer m4uK**
>
> ## Our Response to Weakness 1
> We appreciate the reviewer's emphasis on adhering to established evaluation metrics in quantum chemistry. It's important to note that there isn't a singular metric that fully encapsulates the complexity of this field. Our approach, similar to the ones adopted in FermiNet (Physical Review Research) [11], improved fermionic networks (NIPS2017 Workshop) [12] and PsiFormer (ICLR2023) [13], places considerable importance on computational efficiency as a key metric. Striking a balance between computational cost and accuracy is crucial, especially considering that highly accurate simulations solving the Schrödinger equation without any approximations demand extensive computational resources and time.
>
> To address the concerns about precision, we have included a comprehensive precision analysis in the supplementary, due to space limitations in the main paper. This analysis aligns with the precision levels used in previous state-of-the-art works [14, 15, 16]. For further details on our model's accuracy, particularly for single elements and molecule experiments, we direct the reviewer's attention to Table 7 and Table 8 in appendices due to space limitation. These tables provide a more granular view of our results, demonstrating our commitment to both accuracy and computational efficiency in our research.
>
> ## Our Response to Weakness 2
> We appreciate the reviewer’s suggestion to align our methodology with established terminology in the field. Indeed, the integration of machine learning with DFT data to create force fields is a well-explored area. Our approach, while related, introduces a novel dimension by employing a text-to-vector model to generate embeddings from simulations. This is a distinctive feature compared to methodologies used in other state-of-the-art works such as Equiformer [14], SpookyNet [15], and SchNet [16]. Acknowledging the reviewer’s advice, we will refine our paper to more clearly articulate how our T2E methodology both aligns with and diverges from conventional machine learning force fields. This update will feature a succinct yet comprehensive comparison with existing algorithms, particularly highlighting our innovative use of text-based embeddings. Additionally, we plan to broaden our literature review to more effectively position our work within the established framework of machine learning force fields, ensuring recognition of relevant prior contributions. We are confident that these revisions will underscore the distinctiveness and value of our approach in the field.
>
> ## Our Response to Weakness 3
> Our experimental design aligns with the methodologies of prominent works like PsiFormer and FermiNet. These studies often employ datasets distinct from those in the Open Catalyst Project [17], MD17 [18, 19, 20], or QM9 [21, 22] focusing on reducing computational demands while maintaining reasonable accuracy. Our approach further aims to lower the entry barrier into material science, as demonstrated by our novel text-to-vector method, which simplifies the process of material simulation. We emphasize that, in this field, the necessity for domain-specific datasets is not as pronounced as in theoretical machine learning. Many recent top-tier publications [23, 24] employ varied datasets, yet their findings remain valid due to the consistent underlying physics and common data generation methods like DFT. We will include these considerations and relevant citations in our revised manuscript to provide a comprehensive context for our dataset choice and methodology.
>
> ## Our Response to Weakness 4
> We appreciate the reviewer's point regarding the distinct approaches of NN-VMC and DFT methods. Our benchmarking was intended not to equate the theoretical underpinnings of these methods but to evaluate the efficacy of our T2E model in total energy prediction. The choice to compare with DFT, despite its different approach to solving the Schrödinger equation, was motivated by DFT's prevalent use in material simulations. This benchmark helps us demonstrate T2E's performance in a familiar context to the material science community. We acknowledge the differences in computation methods but assert that comparing the end results—total energy predictions—provides useful insights into T2E's capabilities. We will clarify this rationale to ensure the benchmark's relevance and meaningfulness is well-understood in the manuscript.

---

### Official Review · Reviewer_Ld95 · 2023-10-30

**Soundness:** 3 good
**Presentation:** 4 excellent
**Contribution:** 3 good
**Rating:** 8
**Confidence:** 4

**Summary:**

This paper proposes a new approach called Text-To-Energy (T2E) for predicting total energy of a system. T2E combines text-to-vector encoding with a physics-informed multilayer perceptron (PIMLP). T2E encoder  converts the material name, 3D coordinates, and spin into a vector representation which is then fed into PIMLP for predicting total energy. The PIMLP layer incorporates physics knowledge by using the spin, mass number, and valence electrons as additional features. T2E is evaluated on various elements, small molecules, and untrained materials. It matches or exceeds the accuracy of other methods like FermiNet, PsiFormer, and VMC. Overall, T2E is a strong PoC that leverages pre-trained text encoders and physics informed modules (PIMLP) for predicting total energy of a system. This mix of techniques is novel and hasn't been used in similar machine learning models.

**Strengths:**

1. The use of a pre-trained text encoder to convert basic material inputs into a vector, followed by an PIMLP with physical features, is an innovative approach not explored before.

2. High accuracy with minimal parameters: T2E matches or exceeds SOTA methods like DFT and FermiNet while using far fewer parameters (~52k for PIMLP). This allows fast and accurate predictions with minimal training.

3. A major strength is the low computational scaling and training times of T2E. T2E exhibits constant cost with system size rather than quadratic/exponential scaling for other methods such as DFT.

4. Text encoder allows T2E to work directly from basic material names, coordinates and spin. No need for manually engineering features or preprocessing.

**Weaknesses:**

1. While results are promising, T2E is only trained on a few thousand data points and tested on few handful of systems. Testing on much larger and diverse datasets would be needed for broader adoption.

2.  Ablation studies that remove various parts of T2E could help clarify how important each component is to the model's performance. This is not explored in the current version of the paper. For example, why was the CLIP text encoder chosen as text encoder?

3. T2E is benchmarked on small atoms and molecules. Evaluating performance on slightly larger systems (methane etc) would better validate capabilities.

**Questions:**

1. How would the performance of T2E be impacted by replacing the current CLIP based text encoder with bigger models like GPT-3?

2. Does scaling up the  size of PIMLP  lead to an improvement in model accuracy?

3. Could this approach be generalized to force predictions, potentially making it applicable for tasks like geometry optimization?

---

> ### Author Response · Authors · 2023-11-23
> **T2E's Official Response to Reviewer Ld95**
>
> ## Our Response to Weakness 1
> While our current results with T2E are based on a limited dataset, they serve as a preliminary demonstration of the model's potential. We fully acknowledge that broader testing is crucial for wider adoption and validation of our approach. To this end, we are actively planning to expand our dataset to include a greater variety of molecules and significantly increase the number of data points. This expansion will not only provide more comprehensive testing but also aid in further optimizing and generalizing our model's performance. Our ongoing and future work is firmly committed to exploring these avenues, thereby enhancing the robustness and applicability of T2E in diverse real-world scenarios.
>
> ## Our Response to Weakness 2
>
> We acknowledge the importance of ablation studies in demonstrating the significance of each component in T2E. While these studies were not included in the main paper due to page constraints, we have detailed them in the supplementary materials. Understanding that reviewers are not required to review these materials, we summarize key findings in asnwer to Question 2
>
> ## Our Response to Weakness 3
> Answered this question in response to Question 3
>
> ## Our Response to Question 1
>
> | Element | MAE (eV) (Bert) | MAE (eV) (CLIP) |
> |---------|-----------------|-----------------|
> | Be      | $$1.6374 \times 10^{-5}$$ | $$0.5567 \times 10^{-5}$$ |
> | B       | $$8.3745 \times 10^{-5}$$ | $$10 \times 10^{-5}$$|
> | C       | $$30 \times 10^{-5}$$ | $$10 \times 10^{-5}$$|
>
> *Table 1: Bert vs CLIP MAE comparison. Values are obtained from 5 independent runs.
>
> Our choice to use CLIP in T2E was initially influenced by its widespread adoption and proven effectiveness across various scientific domains. This broad usage has demonstrated CLIP's capability to encode diverse and complex data types, making it particularly suitable for our needs in chemical data encoding. [1, 2, 3, 4, 5, 6]
>
> Furthermore, we compared Bert [7] and CLIP's performances for Be, B, and C. This comparison is reflected in Table 1. We see that CLIP outperforms Bert in 2 of 3 of the materials. However, the difference is not hugely impactful at this scale yet. But we assume it would become effective with larger datasets. In our paper, we've added a note detailing our choice of the CLIP encoder, including its comparison with other encoders for clarity.
>
> ## Our Response to Question 2
>
> Our experiments revealed that increasing the hidden parameters, without altering the physical information, led to overfitting rather than performance improvements. Regularization techniques like $L_1$ and $L_2$ regularization did not yield meaningful benefits. As shown in Figure 8 of the appendices, incorporating additional physical information into the MLP layer significantly boosted accuracy. Specifically, excluding external physical information greatly limited the model’s capabilities. The integration of mass number data aided training stability but did not improve accuracy. However, adding valence electron information alongside mass number markedly enhanced performance on both training and validation sets. Surprisingly, including total electron count did not further benefit the model, likely because this information is implicitly represented in our neutral element-based model. Notably, the role of valence electrons was found to be crucial, as their inclusion significantly improved model performance compared to total electron count.
>
> Furthermore, we observed a decrease in precision when physical information was injected into only one layer, with no substantial improvement seen with more than two injections. These findings underscore the tailored design of T2E, where each component plays a specific role in optimizing the model's performance.
>
> ## Our Response to Question 3
> Indeed, we see promising potential in generalizing our approach to include force predictions, which could extend its applicability to tasks like geometry optimization. Our text-embedding approach lowers the computational cost as well as the need of domain knowledge by simplifying the input data to the model. As part of our future work, we plan to explore this possibility by incorporating and analyzing data from larger molecules. We intend to leverage comprehensive datasets such as the Open Catalyst Project [8] and QM9 [9, 10] for these extensions. This expansion into force predictions is an exciting avenue that could significantly broaden the scope and utility of our method. We believe with additional text information such as surface, molecular dynamic, graph, and phase properties.

---

### Author Response · Authors · 2023-11-23
**T2E's Official Response to Reviewers**

Dear All,

We would like to express our sincere gratitude to the reviewers for their insightful comments, valuable suggestions and dedicated time. We will revise our manuscript with the arguments and the related cites we have provided to the reviewers.

We have included references in here in order to utilize the given limited space efficiently.

## References

[1]: Feighelstein, Marcelo, et al. "Deep learning for video-based automated pain recognition in rabbits." Scientific Reports 13.1 (2023): 14679.

[2]: Défossez, Alexandre, et al. "Decoding speech perception from non-invasive brain recordings." Nature Machine Intelligence (2023): 1-11.

[3]: Ray, Partha Pratim. "ChatGPT and forensic science: a new dawn of investigation." Forensic Science, Medicine and Pathology (2023): 1-2.

[4]: Zhou, Qiongyi, et al. "Exploring the brain-like properties of deep neural networks: a neural encoding perspective." Machine Intelligence Research 19.5 (2022): 439-455.

[5]: Liu, Yulong, et al. "BrainCLIP: Bridging Brain and Visual-Linguistic Representation via CLIP for Generic Natural Visual Stimulus Decoding from fMRI." arXiv preprint arXiv:2302.12971 (2023).

[6]: Yu, Zhuoran. "A Survey on CLIP-Guided Vision-Language Tasks." Highlights in Science, Engineering and Technology 12 (2022): 153-159.

[7]: Devlin, Jacob, et al. "Bert: Pre-training of deep bidirectional transformers for language understanding." arXiv preprint arXiv:1810.04805 (2018).

[8]: Chanussot, Lowik, et al. "Open catalyst 2020 (OC20) dataset and community challenges." Acs Catalysis 11.10 (2021): 6059-6072.

[9]: Ruddigkeit, Lars, et al. "Enumeration of 166 billion organic small molecules in the chemical universe database GDB-17." Journal of chemical information and modeling 52.11 (2012): 2864-2875.

[10]: Ramakrishnan, Raghunathan, et al. "Quantum chemistry structures and properties of 134 kilo molecules." Scientific data 1.1 (2014): 1-7.

[11] Pfau, David, et al. "Ab initio solution of the many-electron Schrödinger equation with deep neural networks." Physical Review Research 2.3 (2020): 033429.

[12] Spencer, James S., et al. "Better, faster fermionic neural networks." arXiv preprint arXiv:2011.07125 (2020).

[13] von Glehn, Ingrid, James S. Spencer, and David Pfau. "A self-attention ansatz for ab-initio quantum chemistry." arXiv preprint arXiv:2211.13672 (2022).

[14] Liao, Yi-Lun, and Tess Smidt. "Equiformer: Equivariant graph attention transformer for 3d atomistic graphs." arXiv preprint arXiv:2206.11990 (2022).

[15] Unke, Oliver T., et al. "SpookyNet: Learning force fields with electronic degrees of freedom and nonlocal effects." Nature communications 12.1 (2021): 7273.

[16] Schütt, Kristof T., et al. "Schnet–a deep learning architecture for molecules and materials." The Journal of Chemical Physics 148.24 (2018).

[17] Chanussot, Lowik, et al. "Open catalyst 2020 (OC20) dataset and community challenges." Acs Catalysis 11.10 (2021): 6059-6072.

[18] Chmiela, Stefan, et al. "Machine learning of accurate energy-conserving molecular force fields." Science advances 3.5 (2017): e1603015.

[19] Chmiela, Stefan, et al. "Towards exact molecular dynamics simulations with machine-learned force fields." Nature communications 9.1 (2018): 3887.

[20] Schütt, Kristof T., et al. "Quantum-chemical insights from deep tensor neural networks." Nature communications 8.1 (2017): 13890.

[21] Ruddigkeit, Lars, et al. "Enumeration of 166 billion organic small molecules in the chemical universe database GDB-17." Journal of chemical information and modeling 52.11 (2012): 2864-2875.

[22] Ramakrishnan, Raghunathan, et al. "Quantum chemistry structures and properties of 134 kilo molecules." Scientific data 1.1 (2014): 1-7.

[23] Fiedler, Lenz, et al. "Predicting electronic structures at any length scale with machine learning." npj Computational Materials 9.1 (2023): 115.

[24] Li, He, et al. "Deep-learning density functional theory Hamiltonian for efficient ab initio electronic-structure calculation." Nature Computational Science 2.6 (2022): 367-377.

[25] Duval, Alexandre Agm, et al. "Faenet: Frame averaging equivariant gnn for materials modeling." International Conference on Machine Learning. PMLR, 2023.

[26] Schütt, Kristof, et al. "Equivariant message passing for the prediction of tensorial properties and molecular spectra." International Conference on Machine Learning. PMLR, 2021.

[27] Sun, Qiming, et al. "PySCF: the Python‐based simulations of chemistry framework." Wiley Interdisciplinary Reviews: Computational Molecular Science 8.1 (2018): e1340.

[28] Press, William H. Numerical recipes 3rd edition: The art of scientific computing. Cambridge university press, 2007.